# Collaborative Multiagent Reinforcement Learning in Homogeneous Swarms

## Abstract

A deep reinforcement learning solution is developed for a collaborative multiagent system. Individual agents choose actions in response to the state of the environment, their own state, and possibly partial information about the state of other agents. Actions are chosen to maximize a collaborative long term discounted reward that encompasses the individual rewards collected by each agent. The paper focuses on developing a scalable approach that applies to large swarms of homogeneous agents. This is accomplished by forcing the policies of all agents to be the same resulting in a constrained formulation in which the experiences of each agent inform the learning process of the whole team, thereby enhancing the sample efficiency of the learning process. A projected coordinate policy gradient descent algorithm is derived to solve the constrained reinforcement learning problem. Experimental evaluations in collaborative navigation, a multi-predator-multi-prey game, and a multiagent survival game show marked improvements relative to methods that do not exploit the policy equivalence that naturally arises in homogeneous swarms.

## 1 Introduction

We consider problems in which groups of agents learn to collaborate. The setup consists of a world, or environment, in which individual agents observe the state of the world, their own state, and possibly some partial information about the state of other agents. Out of the available information agents settle on a choice of action that determines the collection of an instantaneous reward. Our interest is in long term collaborative setups in which the team as a whole acts to maximize the sum of discounted rewards collected by all the members of the group. We restrict attention to homogeneous teams where all agents have identical dynamics and reward structures and, even if our methods work for small number of agents, we focus the effort on large swarms with up to a few hundred agents. Although having a homogeneous team is perhaps the simplest multiagent learning problem we can conceive, the challenges are still significant because the action choices of any agent can have, and most likely will have, some bearing on the actions chosen by other agents. Indeed, agents' actions have a direct effect on the evolution of their own state, the state of the world, and the states of other agents. But this generates an indirect coupling between the actions of different agents as they reason about the collective effects of their individual choices (Eksin et al. (2013a;b)).

Collaborative homogeneous swarms are as common in robotics as they are complex. A somewhat canonical example is a collaborative navigation problem in which a group of $n$ autonomous agents must stake positions in a set of $m$ surveyed points (Berman et al. (2009)). More practical examples are autonomous warehouse management (Enright & Wurman (2011)), collaborative assembly (Knepper et al. (2013)), and concurrent control and communication for teams of robots Stephan et al. (2017). A large literature exists to generate the necessary autonomous collective behavior but the inherent hardness of the problem limits their applicability to relatively small teams (Solovey & Halperin (2016)). Our goal here is to leverage the increasing success of deep neural networks in reinforcement learning (Mnih et al. (2013; 2016); Levine et al. (2015); Pathak et al. (2017); Khan et al. (2018); Gupta et al. (2017)) to develop a generic methodology for learning in the context of large scale collaborative homogeneous swarms.

In principle, we can think of the team as a whole as a system to which we apply some reinforcement learning technique. This is certainly a viable approach with interesting variations of actor-critic models having found particular success (Foerster et al. (2017); Lowe et al. (2017)). However, these

techniques are limited in their applicability to small teams with a few agents. This is not unexpected because the sample inefficiency of conventional reinforcement learning gets compounded with the exponential complexity of having agents learning with respect to each other. To circumvent this limitation we exploit the homogeneity of the team. Under this assumption it is natural to conclude that two different agents faced with the same state and information about the state of other agents must have the same policy. Therefore, we can force the learning procedure to find a common policy for all agents, an observation that constitutes the first contribution of this paper:

**(C1)** We formulate collaborative learning for homogeneous teams as a constrained dynamic program in which different agents are required to find a shared common policy.

In forcing all agents to find a common policy we mitigate the sample inefficiency of reinforcement learning as the experiences of any given agent inform the learning process of all other agents. The situation is not unlike meta-learning where we cross-pollinate different tasks to effectively enlarge the size of the training set (Al-Shedivat et al. (2017); Finn et al. (2017)). This similarity of purpose notwithstanding the differences are sufficiently important to warrant the development of a custom learning technique. We do so by working on a variation of the policy gradient method which we adapt to solve the collaborative learning constrained dynamic program mentioned in (C1).

Interestingly, the equivalence between policies can be incorporated into the problem formulation in several different ways (see Section 2). Our experimental results show that the most convenient formulation is one in which we introduce a (fictitious) common policy and require all agents to find policies that agree with this common policy. In order to solve this optimization we utilize a coordinate projected policy gradient algorithm whose development is the second contribution of this paper:

**(C2)** We develop a coordinate projected policy gradient algorithm. At each step of this iterative algorithm agents roll out trajectories according to the common policy and utilize the experience to update their individual policies. This is a policy gradient descent step along individual agent coordinates. After this update we roll out trajectories in which individual policies are pitched out against the common centralized policy. This is used to implement a policy gradient descent step on the central policy. After these two separate coordinate descent steps we project the disparate policies into a common policy and proceed to the next iteration.

We provide experimental evaluations in a collaborative navigation problem and a multi-predator-multi-prey game (from Lowe et al. (2017)), and a multiagent survival game (from Zheng et al. (2017). The numerical experiments show marked improvements in rewards relative to methods that do not force policy equivalence (Lowe et al. (2017)).

## 1.1 RELATED WORK

Multiagent reinforcement learning problems have a long history Littman (1994); Hu & Wellman (2003); Busoniu et al. (2006); Conitzer & Sandholm (2007). Early approaches focused on tabular methods to compute Q-values for Markov games (e.g, Hu & Wellman (2003)) and developed several solution methodologies for finding competitive and collaborative equilibrium (e.g., Conitzer & Sandholm (2007)). Building on the success of deep reinforcement learning there has been a recent buildup of interest in using high capacity neural network models. These include the use of neural networks in two-player games (Tampuu et al. (2017)) and in generic Bayesian games (Da Silva et al. (2006); Eksin et al. (2013a;b); Hong et al. (2018)) in which each of the agents must build a model for the behavior of other agents. In terms of novel algorithmic development, (Foerster et al. (2017)) and (Lowe et al. (2017)) propose a variation of actor-critic methods in which each agent is modeled as a decentralized actor working in conjunction with a centralized critic with parameter sharing among the agents. In Bayesian games formulations as well as in actor-critic approaches, it is possible to have a heterogeneous agent mix but it is difficult to scale the number of agents. This is different from our interest which is on leveraging homogeneity to provide scalability to large swarms. An alternative approach to achieve scalability is to work with mean field game models in which we assume the number of agents to be infinite and train policies that can be deployed on individual robots while operating on summary statistics of the team's state (Mguni et al. (2018); Yang et al. (2017)). Our work differs in that we operate on information about the state of individual agents which can be incomplete and typically involves a few agents only.

## 2 Markov Collaborative Reinforcement Learning

We consider policy learning problems in a collaborative Markov team (Littman (1994)). The team is composed of $N$ agents generically indexed by $n$ which at any given point in time $t$ occupy a position $x_{nt} \in \mathcal{X}$ in configuration space and must choose an action $a_{nt} \in \mathcal{A}$ in action space. Agents operate in a world, or environment, whose state at time $t$ we denote as $w_t$. The team and environment are assumed Markov so that if we collect all agents' configurations in the vector $\mathbf{x}_t := [x_{1t}; \ldots; x_{Nt}]$ and all actions in the vector $\mathbf{a}_t := [a_{1t}; \ldots; a_{Nt}] \in \mathcal{A}^N$ the evolution of the system is completely determined by the conditional transition probability $p\left(\mathbf{x}_{t+1}, w_{t+1} \mid \mathbf{x}_t, \mathbf{a}_t, w_t\right)$. We further assume that agents are statistically identical in that the probability transition kernel is invariant to agent permutations. This implies the transition dynamics are the same for all agents so that if we swap two of them in configuration and action space we expect to see the same statistical evolution. The assumption is justified because the robotic swarm is assumed homogeneous.

In order to choose their actions $a_{nt}$, agents have access to their own states $x_{nt}$, the state of the world $w_t$ and some possibly partial information $I(\mathbf{x}_{-nt})$ about the state of other agents $\mathbf{x}_{-nt} = [x_{mt}]_{m \neq n}$. All of these variables are grouped in the local state $s_{nt} = (x_{nt}, w_t, I(\mathbf{x}_{-nt}))$. The action $a_{nt}$ is chosen to be Markov with respect to this state. Therefore, the policy of agent $n$ is a function $\pi_n$ that chooses actions according to

$$a_{nt} \;=\; \pi_n(s_{nt}) \;=\; \pi_n(x_{nt}, w_t, I(\mathbf{x}_{-nt})). \tag{1}$$

Observe that the policy is chosen to be Markov even though there may be advantages in keeping track of past states and past information. The *team* is Markov but from the perspective of an individual agent the evolution of the system is not necessarily Markov unless the information $I(\mathbf{x}_{-nt})$ is a complete description of the states of other agents. For future reference we define $\boldsymbol{\pi} := [\pi_1; \ldots; \pi_N]$ to group the policies of all agents and $\boldsymbol{\pi}_{-n} = [\pi_m]_{m \neq n}$ to group the policies of all agents except $n$.

As agents operate in their environment, they collect *individual* rewards $r_n(\mathbf{x}_t, w_t, a_{nt})$ which depend on the configuration of the team $\mathbf{x}_t$, the state of the world $w_t$ and their own individual action $a_{nt}$. The quantity of interest to agent $n$ is not this instantaneous reward but rather the long term reward accumulated over a time horizon $T$ as discounted by a factor $\gamma$,

$$R_n := \sum_{t=0}^{T} \gamma^t r_n(\mathbf{x}_t, w_t, a_{nt}). \tag{2}$$

The reward $R_n$ in equation 2 is stochastic as it depends on the trajectory's realization. In conventional reinforcement learning, agent $n$ would define the cost $\tilde{L}_n(\pi_n) := \mathbb{E}_{\pi_n}(R_n)$ and search for a policy $\pi_n$ that maximizes this long term expected reward. Naturally, the expectation $\mathbb{E}_{\pi_n}(R_n)$ implicitly depends on the policies of other agents since this would affect the transition probability $p\left(\mathbf{x}_{t+1}, w_{t+1} \mid \mathbf{x}_t, \mathbf{a}_t, w_t\right)$. To emphasize this fact we write $\tilde{L}_n(\pi_n) = \mathbb{E}_{\pi_n, \boldsymbol{\pi}_{-n}}(R_n)$ where, we recall, $\boldsymbol{\pi}_{-n} = [\pi_m]_{m \neq n}$ represents the policies of all agents except $n$. Since we are interested in collaborative teams, we consider a different formulation in which agents strive to maximize the sum of accumulated rewards across all members of the team

$$\mathbb{E}_{\boldsymbol{\pi}}\left[\sum_{n=1}^{N} R_n\right] \;=\; \sum_{n=1}^{N} \mathbb{E}_{\pi_n, \boldsymbol{\pi}_{-n}}[R_n] \;:=\; \sum_{n=1}^{N} L_n(\pi_n, \boldsymbol{\pi}_{-n}), \tag{3}$$

where we have defined $L_n(\pi_n, \boldsymbol{\pi}_{-n}) := \mathbb{E}_{\pi_n, \boldsymbol{\pi}_{-n}}[R_n]$ which is the component of the reward that is collected by agent $n$. To clarify ideas we discuss an example.

**Example 1 (Collaborative Navigation)** To illustrate the problem formulation consider a collaborative navigation task whereby a team of $N$ agents are tasked with reaching $N$ preassigned goals; see Figure 2 (left). The goals are fixed and we denote their locations as $y_n$. The agents move to approach these goals and at each point in time $t$, the position of agent $n$ is denoted as $x_{nt} \in \mathbb{R}^2$. Agent $n$ observes its location in space perfectly. This is the agent's position in configuration space [cf. equation 1]. Agent $n$ also observes perfectly the location of all goals. Thus, the state of the world available to all agents is $w_t = [y_n; \ldots; y_n]$. Observe that this is an static state of the world. A time varying world state is obtained if the goals are moving around. In addition to this, agent $n$ also observes the positions of other agents relative to its own position. I.e., the information $I(\mathbf{x}_{-nt})$

is a collection of relative locations $x_{mt} - x_{nt}$ for all $m \neq n$. This is a full information example. Partial information is obtained if the location of some agents is unknown or if the location of the agents is known with some error. The reward collected by agent $n$ varies inversely with the distance $\|x_{nt} - y_n\|$ to its assigned target and incorporate a negative reward for colliding with other agents. We wish to learn a policy that maximizes the sum of accumulated rewards across all members of the team [cf. equation 3]. We expect that this will induce a behavior in which all agents move to their assigned targets while avoiding collisions with each other.

## 2.1 Optimal Common Policies

In the loss in equation 3 agent $n$ collects rewards $r_n(\mathbf{x}_t, w_t, a_{nt})$ whose expectation with respect to the joint policy $\boldsymbol{\pi}$ is $L_n(\pi_n, \boldsymbol{\pi}_{-n})$. The team acts as a group to make the expected cumulative reward $\sum_{n=1}^{N} L_n(\pi_n, \boldsymbol{\pi}_{-n})$ as large as possible. This would call for finding the policy $\boldsymbol{\pi}^\dagger = \arg\max_{\boldsymbol{\pi}} \sum_{n=1}^{N} L_n(\pi_n, \boldsymbol{\pi}_{-n})$. While this is indeed a sensible definition for an optimal policy, it requires learning separate policies for each individual agent. This is intractable for large $N$, motivating a restriction in which all agents are required to execute a common policy,

$$\pi_n^* := \arg\max \sum_{n=1}^{N} L_n(\pi_n, \boldsymbol{\pi}_{-n}), \qquad \text{s.t. } \pi_n = \pi_m, \text{ for all } n \neq m. \qquad (4)$$

The multi-agent reinforcement learning formulation in equation 4 takes advantage of the fact that agents are statistically identical to simplify the learning space. We emphasize that the policies $\pi_n^*$ in equation 5 can be different from the individual policies that compose the joint policy $\boldsymbol{\pi}^\dagger = \arg\max_{\boldsymbol{\pi}} \sum_{n=1}^{N} L_n(\pi_n, \boldsymbol{\pi}_{-n})$ even when agents are statistically identical as it may be beneficial for different agents to take different actions when faced with the same state.

The formulation in equation 4 can be further simplified with the definition of a common policy $\pi$. When we do this, we can replace the $N^2$ constraints $\pi_n = \pi_m$ by the $N$ constraints $\pi_n = \pi$. More importantly, the policy $\boldsymbol{\pi}_{-n}$ of other agents that appears as an argument in the loss $L_n(\pi_n, \boldsymbol{\pi}_{-n})$ can be replaced by a policy $\boldsymbol{\pi}_{-n}$ in which $\pi_m = \pi$ for all $m \neq n$. Denoting the resulting loss as $L_n(\pi_n, \boldsymbol{\pi}_{-n}) = L_n(\pi_n, \pi)$ we can reformulate equation 4 as

$$(\pi_n^*, \pi^*) = \arg\max \sum_{n=1}^{N} L_n(\pi_n, \pi), \qquad \text{s.t. } \pi_n = \pi \text{ for all } n. \qquad (5)$$

In the spirit of having simpler problem formulations, we can eliminate the individual policies altogether. Given that the optimal policies are such that $\pi_n^* = \pi^*$ for all $n$ we can simply replace the $L_n(\pi_n, \pi)$ by $L_n(\pi) := L_n(\pi, \pi)$ and remove the constraints to write

$$\pi^* = \arg\max \sum_{n=1}^{N} L_n(\pi) \qquad (6)$$

The problem formulations in equation 4, equation 5, and equation 6 are all equivalent. However, this doesn't imply that algorithms to solve them are equivalent. Our experiments have shown that a projected gradient descent algorithm working on equation 5 is most effective. Therefore, the purpose of this paper is to develop a policy gradient algorithm for solving equation 5.

## 3 Projected Coordinate Policy Gradient

Let us reiterate the problem in equation 4 in terms of the parameterization of the policy. Eqn 4 can be interpreted as a problem where we aim to solve is to find the best set of parameters $\theta^*$ that parameterizes a policy $\pi_\theta$ to maximize the sum of rewards $R_i$ for all agents over some time horizon $T$. Thus parametrized version of equation 4 can be written as :

$$(\theta_n^*) = \arg\max \sum_{n=1}^{N} L_n(\theta_n, \boldsymbol{\theta_{-n}}), \qquad \text{s.t. } \theta_n = \theta_m \text{ for all } n \neq m. \qquad (7)$$

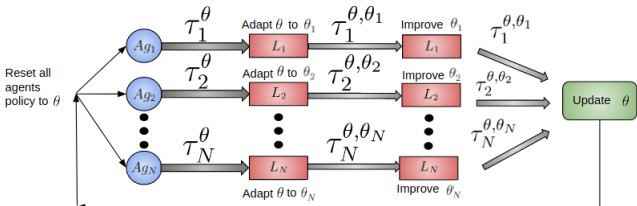

Figure 1: **Distributed Multi-Agent Policy Gradients:** Each agent $n$ ($Ag_n$) starts under policy parametrized by $\theta$ and uses it to collect experience $\tau_n^\theta$. $\tau_n^\theta$ is used to minimize agent $Ag_n$'s loss function $L_n$ and adapt its policy from $\theta$ to $\theta_n$. Now, $Ag_n$ uses policy parametrized by $\theta_n$ assuming other agents policies remain $\theta$. The trajectory generated in this case is denoted by $\tau_n^{\theta,\theta_n}$ and is used to improve $Ag_n$'s policy by taking gradients w.r.t this intermediate policy. Finally, using this improved policy, we collect another new trajectory $\tau_n^{\theta,\theta_n}$. These new trajectories are used to update $\theta$.

However, as stated above this problem can be intractable for large $N$. Rewriting the parametrized version of the more tractable optimization in Eqn 5 we get:

$$(\theta_n^*, \theta^*) = \arg\max \sum_{n=1}^{N} L_n(\theta_n, \theta), \qquad \text{s.\,t.}\, \theta_n = \theta \text{ for all } n. \tag{8}$$

The difference between Eqn 7 and Eqn 8 is that we have formed $N$ copies of $\theta$ labeled $\theta_n$ and put a constraint that $\theta = \theta_n$. This approach allows us to look at the problem in a different light. Similar to other distributed optimization problems such as ADMM Boyd et al. (2011), we can decouple the optimization over $\theta_n$ from that of $\theta$. The general approach is an iterative process where

1. For each agent $n$, optimize the corresponding $\theta_n$

2. Consolidate the $\theta_n$ into $\theta$

This is often realized as a projected gradient descent where for each agent $n$, we apply the gradients $\theta_n \leftarrow \theta_n + \alpha_1 \nabla_{\theta_n} L(\theta, \theta_n)$ as well as applying a gradient $\theta \leftarrow \theta + \alpha_2 \nabla_\theta \sum_{n=1}^{N} L(\theta, \theta_n)$. Then, in the next iteration all agents start at $\theta_n$ where $\theta_n$ is realized by taking a projection step such that $\theta_n = \theta \leftarrow \frac{1}{N+1}(\theta + \sum_{n=1}^{N} \theta_n)$ is taken to satisfy the constraint in equation 8. However, when computing this projected gradient step, we need to keep track of all $\theta_n$ to compute the average. This is infeasible if this is done for a large number of agents. Instead a simple approximation to the projected gradient is used by setting $\theta_n \leftarrow \theta$. In the next subsection, we present our algorithm *Distributed Multi Agent Policy Gradient* or DiMA-PG and its practical implementation.

### 3.1 DISTRIBUTED MULTI-AGENT POLICY GRADIENTS (DIMA-PG)

In this section, we propose the *Distributed Multi Agent Policy Gradient* (DiMA-PG) algorithm which learns a centralized policy that can be deployed across all agents. Consider a population $Pop$ from which $N$ statistically identical agents are sampled according to a distribution $P(Pop)$. The parameters $\theta_n$ of this agent-specific policy are updated by taking the gradient w.r.t $\theta$ at the specific value of $\theta = \theta_0$ (where $\theta_0$ is your current central (or common) policy):

$$\theta_n \leftarrow \theta_0 + \alpha_1 \nabla_{\theta_n} L_n(\theta_n, \theta)|_{\theta=\theta_0, \theta_n=\theta_0} \tag{9}$$

where $\alpha$ is step size hyperparameter and $L(\theta, \theta_n)$ is as defined in Eqn 2. Note that $L(\theta_0, \theta_0)$ is when all agents follow policies $\pi_{\theta_0}$ while $L(\theta_0, \theta_n)$ is when agent $n$ follows $\pi_{\theta_n}$ and all other agents follow $\pi_{\theta_0}$. We do this because, when the environment is held constant w.r.t agent, then the problem for agent $n$ reduces to a MDP Sutton & Barto (1998).

In practice, we can take $k$ gradient steps instead of just one as presented in Eqn 9. This can be done with the following inductive steps

$$\theta_n^{[0]} = \theta_0$$
$$\theta_n^{[k]} = \theta_n^{[k-1]} + \alpha_2 \nabla_{\theta_n} L_n(\theta, \theta_n)|_{\theta=\theta_0, \theta_n=\theta_n^{[k-1]}} \tag{10}$$
$$\theta_n = \theta_n^{[k]}$$

Finally, we update $\theta$:

$$\theta \leftarrow \theta + \epsilon \nabla_\theta \sum_{n=1}^N L_n(\theta_n, \theta) \tag{11}$$

Numerically, we approximate $\nabla_{\theta_n} L_n(\theta_n, \theta)$ by drawing $l$ trajectories where agent $n$ uses policy $\pi_{\theta_n}$ while all other agents uses policy $\pi_\theta$ and averaging over the policy gradients that each trajectory provides (Williams (1992); Sutton & Barto (1998)).

Let us define the trajectory $\tau_n^\theta$ when all agents are following policy $\pi_\theta$

$$\tau_n^\theta = \left\{ [s_n^{t_0,\theta}, a_n^{t_0,\theta}, a_{1,\ldots N \neq n}^{t_0,\theta}, r_n^{t_0}], [s_n^{t_1,\theta}, a_n^{t_1,\theta}, a_{1,\ldots N \neq n}^{t_1,\theta}, r_n^{t_1}] \ldots, [s_n^{t_T,\theta}, a_n^{t_T,\theta}, a_{1,\ldots N \neq n}^{t_T,\theta}, r_n^{t_T}] \right\} \tag{12}$$

and $\tau_n^{\theta,\theta_n}$ to be the trajectory when agent $n$ follows policy $\pi_{\theta_n}$ and all other agents follow policy $\pi_\theta$.

$$\tau_n^{\theta,\theta_n} = \left\{ [s_n^{t_0,\theta_n}, a_n^{t_0,\theta_n}, a_{1,\ldots N \neq n}^{t_0,\theta}, r_n^{t_0}], [s_n^{t_1,\theta_n}, a_n^{t_1,\theta_n}, a_{1,\ldots N \neq n}^{t_1,\theta}, r_n^{t_1}], \right.$$
$$\left. \ldots, [s_n^{t_T,\theta_n}, a_n^{t_T,\theta_n}, a_{1,\ldots N \neq n}^{t_T,\theta}, r_n^{t_T}] \right\} \tag{13}$$

The trajectories $\tau_n^\theta$ and $\tau_n^{\theta,\theta_n}$ are random variables drawn from distributions $P_n(\tau_n^\theta|\theta)$ and $P_n(\tau_n^{\theta,\theta_n}|\theta, \theta_n)$ respectively. The individual agent policy parameters, $\theta_n$ are also random variables with distribution $P_n(\theta_n|\theta)$. The overall optimization can be written as:

$$\max_\theta \mathbb{E}_{n \sim P(Pop)} \left[ \mathbb{E}_{\tau_n^\theta \sim P_n(\tau_n^\theta|\theta)} \left[ \mathbb{E}_{\tau_n^{\theta,\theta_n} \sim P_n(\tau_n^{\theta,\theta_n}|\theta,\theta_n)} [L_n(\theta_n, \theta)|(\tau_n^\theta, \theta)] \right] \right] \tag{14}$$

Assuming, we sample N agents, Eqn. 14 can be rewritten as:

$$\max_\theta \frac{1}{N} \sum_{n=1}^N \left[ \mathbb{E}_{\tau_n^\theta \sim P_n(\tau_n^\theta|\theta)} \left[ \mathbb{E}_{\tau_n^{\theta,\theta_n} \sim P_n(\tau_n^{\theta,\theta_n}|\theta,\theta_n)} [L_n(\theta_n, \theta)|(\tau_n^\theta, \theta)] \right] \right] \tag{15}$$

To learn $\theta$, we use policy gradient methods (Williams (1992); Sutton & Barto (1998)) which operate by taking the gradient of Eqn. 15. One can also use recently proposed state of the art methods for policy gradient methods (Schulman et al. (2015b;a)). The gradient for each agent in Eqn 15 (the quantity inside the sum) w.r.t $\theta$ can be written as:

$$\nabla_\theta \mathcal{L}_n(\theta, \theta_n) = \mathbb{E}_{\tau_n^\theta \sim P_n(.|\theta), \tau_n^{\theta,\theta_n} \sim P_n(.|\theta,\theta_n)} \left[ L_n(\theta_n, \theta) \nabla_\theta \log \pi_{\theta_n}(\tau_n^{\theta,\theta_n}) + L_n(\theta_n, \theta) \nabla_{\theta_n} \log \pi_\theta(\tau_n^\theta) \right] \tag{16}$$

The policy gradient for each agent consists of two policy gradient terms, one over the trajectories $\tau_n^{\theta,\theta_n}$ sampled using $(\theta, \theta_n)$ and another term over the trajectories $\tau_n^\theta$ sampled using $\theta$. It may be noted that the terms from the agent specific policy improvement when the other agents are held stationary (Eqn 10) do not appear in the final term. We show that it is possible to marginalize these terms out in the derivation for the gradient and point the reader to the appendix for a full derivation of the policy gradient. The full algorithm for DiMA-PG is presented in Algorithm 1.

## 4 EXPERIMENTS

### 4.1 ENVIRONMENTS

To test the effectiveness of DIMAPG, we perform experiments on both collaborative and competitive tasks. The environments from (Lowe et al. (2017)) and the many-agent (MAgent) environment from

---

**Algorithm 1** Distributed Multi Agent with Policy Gradients (DIMA-PG)

---

**Require:** Initial random central policy $\theta$, step-size hyperparameters $\alpha_1, \alpha_2, \epsilon$ and distribution over agent population P(Pop)

1: **while** True **do**
2:   Sample $N$ agents $\sim$ P(Pop)
3:   **for** all agents **do**
4:     Collect trajectory $\tau_n^\theta$ as given in Eqn 12 and evaluate agent loss $L_n(\theta_n, \theta)|_{\theta=\theta_0, \theta_n=\theta_0}$
5:     Compute agent specific policy $\theta_n$ according to Eqn 9
6:     Using $\theta$ and $\theta_n$ compute trajectory $\tau_{\theta, \theta_n}$ according to Eqn 13
7:   **end for**
8:   Compute policy gradient $\nabla_\theta L_n(\theta_n, \theta)$ for every agent according to Eqn 16
9:   Update central policy $\theta \leftarrow \theta + \epsilon \nabla_\theta \sum_{n=1}^{N} L_n(\theta_n, \theta)$ (Eqn 11)
10: **end while**

---

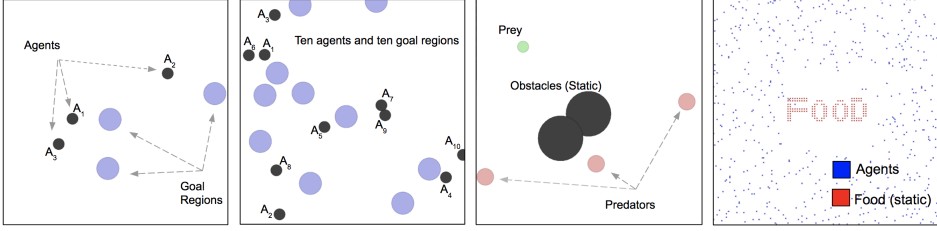

Figure 2: **Multi-agent environments for testing:** We consider both collaborative as well as competitive environments. **Left:** Collaborative Navigation (with 3 agents) **Center Left:** Collaborative Navigation for 10 agents. **Center Right:** Predator-Prey **Right:** Survival with many (630) agents

(Zheng et al. (2017)) are adapted for our experiments. We setup the following experiments to test out our algorithm :

**Collaborative Navigation** This task consists of $N$ agents and $N$ goals. All agents are identical, and each agent observes the position of the goals and the other agents relative to its own position. The agents are collectively rewarded based on the how far any agent is from each goal. Further, the agents get negative reward for colliding with other agents. This can be seen as a coverage task where all agents must learn to cover all goals without colliding into each other. We test increasing the number of agents and goal regions and report the minimum reward across all agents.

**Predator Prey** This task environment consists of two populations - predators and preys. Prey are faster than the predators. The environment is also populated with static obstacles that the agents must learn to avoid or use to their advantage. All agents observe relative positions and velocities of other agents and the positions of the static obstacles. Predators are rewarded positively when they collide with the preys and the preys are rewarded are negatively.

**Survival** This task consists of a large number of agents operating in an environment with limited resources or food. Agents get reward for eating food but also get reward for killing other agents (reward for eating food is higher). Agents must either rush to get reward from eating food or monopolize the food by killing other agents. However, when the agents kill other agents they incur a small negative reward. Each agent's observations consists of a spatial local view component and a non spatial component. The local view component encodes information about other agents within a range while the non spatial component encodes features such as the agents ID, last action executed, last reward and the relative position of the agent in the environment.

## 4.2 EXPERIMENTAL RESULTS

For all experiments, we use a neural network policy that consists of two hidden layers with 100 units each and uses ReLU nonlinearity. For the collaborative navigation task, we use the vanilla policy gradient or REINFORCE (Williams (1992)) to compute updates ($\theta_n$) and TRPO (Schulman et al. (2015a)) to compute $\theta$. For the Predator Prey and Survival tasks we switch to using REINFORCE for

both $\theta$ and $\theta_n$. To establish baselines, we compare against both centralized and decentralized deep MARL approaches. For decentralized learning, we use MADDPG from (Lowe et al. (2017)) using the online implementation open sourced by the authors. Since the authors in Lowe et al. (2017) already show MADDPG agents work better than other methods on the exact same environments that we are using, where individual agents are trained by DDPG, REINFORCE, Actor-Critic, TRPO, DQN, we do not re implement those algorithms. Instead, we implement a centralized A3C (Actor-Critic) (Mnih et al. (2016)) and centralized TRPO that take in as input the joint space of all agents observations and output actions over the joint space of all agents. We call this the *Kitchensink* approach. Details about the policy architecture for *A3C_Kitchenshink* and *TRPO_Kitchenshink* are provided in the appendix.

### 4.2.1 COLLABORATIVE NAVIGATION

We setup collaborative navigation as described in Section 4.1. Agents are rewarded for being close to the goals (negative square of distance to the goals) and get negatively rewarded for colliding into each other or when they step out of the environment boundary. We also observe that in order to stabilize training, we need to clip our rewards in the range [-1,1]. We use a horizon $T = 200$ after which episodes are terminated. Additional hyper parameters are provided in the Appendix.

We run our proposed algorithm and baselines on this environment when number of agents $n = 3$ and $n = 10$. Since the baselines A3C_Kitchenshink and TRPO_Kitchenshink operate over the joint space, they are setup to maximize the minimum reward across all agents. To have a fair comparison among baselines, we report the minimum reward across all agents. The training curve for our tasks can be seen in Fig 3. We notice that for the simple case, A3C_Kitchenshink performs very well and quickly converges. This is expected since the number of agents is low and the dimensionality of the input space is not large. TRPO_Kitchenshink and MADDPG perform worse and while they converge, the convergence is only seen after 300-400k episodes. When $n$ is increased to ten, we observe that only DIMAPG is able to quickly learn policies for all agents.

|  | *n=3* | n=10 |
|---|---|---|
| Using $\theta$ | -34.8 | -8 |
| Using $\theta_i'$ | -37.19 | -8.5 |
| Fine Tune | -44.17 | -56.3 |

Table 1: **Min. reward across all agents after training (avg. over 100 episodes)**

In our initial hypothesis, we sought to use $\theta$ across all agents since we assumed that the policies for all agents in a given population live close to each other in parameter space. We observe from Table 1 that after training using $\theta$ or $\theta_i'$ (after k-shot adaptation from $\theta$) yields almost similar results thus, verifying our hypothesis. We also consider the case where we train only 1 agent and then fine tune the same policy across all agents. We observe that this yields poor results when the number of agents are increased and is indicated by the "Fine Tune" entry in Table 1

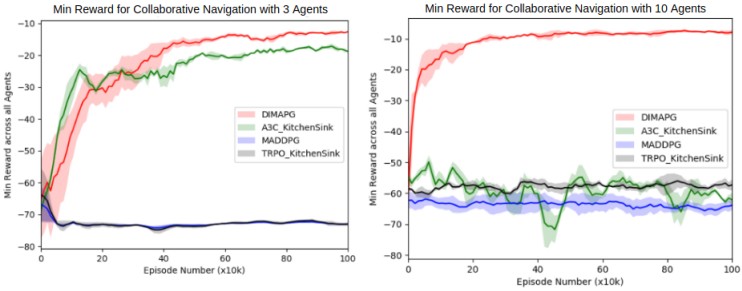

Figure 3: **Min reward vs. number of episodes for Collaborative Navigation:** DIMAPG converges quickly in both scenarios. The protocol followed in the plots involves 5 independent runs for each algorithm with different seeds, darker line represents the mean and the shaded lighter region represents the variance.

### 4.3 PREDATOR PREY

The goal of this experiment is to compare the effectiveness of DIMAPG on competitive tasks. In this task, there exist 2 populations of agents; predators and preys. Extending our hypothesis to this

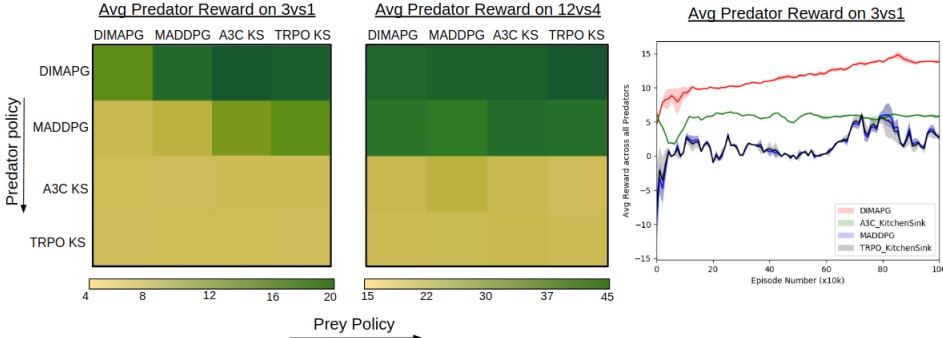

Figure 4: **Results on Predator Prey. Left, Center:** Average predator reward collected over 100 episodes after training different policies for predators and preys. In the 3 Predators vs 1 Prey game, the prey is 30% faster than the predators. In the 12 Predators vs 4 Prey, the prey is 50% faster than the predators. **Right:** Avg predator reward vs episodes during training for 3vs1 game.

task, we would like to learn a single policy for all predators and a single policy for all preys. It is important to note that even though, the policies are different, they are trained in parallel which in the centralized setup enables us to condition each agents trajectory on the actions of other agents even if they are in a different population. We experiment with two scenarios; 12vs1 and 3vs1 predator prey games where the prey are faster than the predator. The horizon used is $T = 200$.

Our results are presented in Fig 4. We observe that DIMAPG is able to effectively learn better policies than both MADDPG and the centralized Kitchensink methods on this competitive task. Similar results with DIMAPG are achieved even when the number of predators and preys are increased.

### 4.4 SURVIVAL

The goal of this experiment is to demonstrate the effectiveness of DIMAPG on environments with a large number of agents. The environment is populated with agents and food (the food is static particles at the center). Agents must learn to survive by eating food. To do so they can either rush to gather food and get reward or monopolize the food by first killing other agents (killing other agents results in a small negative reward). We use DIMAPG to learn the central policy that is deployed across all agents by randomly sampling $N$ agents from the population. We roll out each episode for a horizon of $T = 200$. Each environment is populated with 160 food particles (eating one food particle yields a reward of +5). For this task, it is infeasible to train the other baselines and hence we do not benchmark for this experiment.

| Statistics | N=230 | N=630 |
|---|---|---|
| Food Left | 0 | 0 |
| Survivors | 227 | 490 |
| Average Reward | 946 | 674 |

Table 2: **Statistics on Survival collected over over 100 games using DIMAPG, after training.** Initial average reward for $N = 630$ is -3800 and for $N = 230$ it is -1530.

We gauge the performance of DIMAPG on this task by evaluating the number of surviving agents and the food left at the end of the episode as well as the average reward over agents per episode.(Table 2). It is observed in the case when $N = 225$, the agents do not kill each other and instead learn to gather food. When the number of agents is increased to $N = 630$ agents close to the food rush in to gather food while those further away start killing other agents.

## 5 CONCLUSION AND OUTLOOK

Thus, in this work we have proposed a distributed optimization setup for multi-agent reinforcement learning that learns to combine information from all agents into a single policy that works well for

large populations of homogeneous agents. We show that our proposed algorithm performs better than other state of the art deep multi agent reinforcement learning algorithms when the number of agents are increased. In future work, we intend to explore the idea of learning a policy for teams of heterogeneous agents.

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

**APPENDIX**

## A  DERIVATION FOR MULTI-AGENT POLICY GRADIENT

Following Section 3.1, the overall optimization problem for distributed multi agent reinforcement learning was given as :

$$\max_\theta \mathbb{E}_{n \sim P(Pop)} \left[ \mathbb{E}_{\tau_n^\theta \sim P_n(\tau_n^\theta | \theta)} \left[ \mathbb{E}_{\tau_n^{\theta,\theta_n} \sim P_n(\tau_n^{\theta,\theta_n} | \theta, \theta_n)} [L_n(\theta_n, \theta) | (\tau_n^\theta, \theta)] \right] \right] \tag{17}$$

where trajectories $\tau_n^\theta$ and $\tau_n^{\theta,\theta_n}$ are random variables with distributions $P_n(\tau_n^\theta | \theta)$ and $P_n(\tau_n^{\theta,\theta_n} | \theta, \theta_n)$ respectively. Assuming, we sample N agents, the above Eqn 17 can be rewritten as:

$$\max_\theta \frac{1}{N} \sum_{n=1}^N \left[ \mathbb{E}_{\tau_n^\theta \sim P_n(\tau_n^\theta | \theta)} \left[ \mathbb{E}_{\tau_n^{\theta,\theta_n} \sim P_n(\tau_n^{\theta,\theta_n} | \theta, \theta_n)} [L_n(\theta_n, \theta) | (\tau_n^\theta, \theta)] \right] \right] \tag{18}$$

Let :

$$\mathcal{L}_n(\theta_n, \theta) = \left[ \mathbb{E}_{\tau_n^\theta \sim P_n(\tau_n^\theta | \theta)} \left[ \mathbb{E}_{\tau_n^{\theta,\theta_n} \sim P_n(\tau_n^{\theta,\theta_n} | \theta, \theta_n)} [L_n(\theta_n, \theta) | (\tau_n^\theta, \theta)] \right] \right] \tag{19}$$

Since it is required that we maximize only over theta, we are interested in marginalizing $\theta_n$. Expanding all expectations we can write:

$$\mathcal{L}_n(\theta_n, \theta) = \int \int \int L_n(\theta_n, \theta) P_n(\tau_n^{\theta,\theta_n} | (\theta, \theta_n)) P_n(\theta_n | \theta, \tau_n^\theta) P_n(\tau_n^\theta | \theta) d\tau_n^\theta d\tau_n^{\theta,\theta_n} d\theta_n \tag{20}$$

Assuming, we use the k gradient steps instead of just one as presented in Eqn 10 in the main paper, this can be rewritten as :

$$\mathcal{L}_n(\theta_n, \theta) = \int L_n(\theta_n, \theta) P_n(\tau_n^{\theta,\theta_n} | (\theta, \theta_n)) P_n(\theta_n^{[k]} | \theta_n^{[k-1]}, \tau_n^{\theta_n^{[k-1]}}) P_n(\theta_n^{[k-1]} | \theta_n^{[k-2]}, \tau_n^{\theta_n^{[k-2]}}) \ldots$$
$$P_n(\theta_n^{[1]} | \theta_n^{[0]}, \tau_n^{\theta_n^{[0]}}) P_n(\tau_n^\theta | \theta) d\tau_n^\theta d\tau_n^{\theta,\theta_n^{[0]}} d\tau_n^{\theta,\theta_n^{[1]}} \ldots d\tau_n^{\theta,\theta_n^{[k]}} d\theta_n \tag{21}$$

The term $P_n(\theta_n | \theta, \tau_n^\theta) d\theta_n$ in the above Eqn 20 can be integrated out if we assume a delta distribution for $P_n(\theta_n | \theta, \tau_n^\theta)$:

$$P_n(\theta_n | \theta, \tau_n^\theta) = \delta \left( \theta_0 + \alpha_1 \nabla_{\theta_n} L_n(\theta_n, \theta) |_{\theta = \theta_0, \theta_n = \theta_0} \right) \tag{22}$$

A similar observation can be made for the intermediate terms $P_n(\theta_n^{[1]} | \theta_n^{[0]}, \tau_n^{\theta_n^{[0]}})$, $P_n(\theta_n^{[2]} | \theta_n^{[1]}, \tau_n^{\theta_n^{[1]}})$, ..., $P_n(\theta_n^{[k]} | \theta_n^{[k-1]}, \tau_n^{\theta_n^{[k-1]}})$ in the above Eqn 21. Thus after integrating these terms out (in the above Eqn 20 or 21, we are left with:

$$\mathcal{L}_n(\theta_n, \theta) = \int \int L_n(\theta_n, \theta) P_n(\tau_n^{\theta,\theta_n} | (\theta, \theta_n)) P_n(\tau_n^\theta | \theta) d\tau_n^\theta d\tau_n^{\theta,\theta_n} \tag{23}$$

Taking the gradient of this above equation 23, rewriting the probability distributions in the policy form and rewriting it as an expectation form we get:

$$\nabla_\theta \mathcal{L}_n(\theta_n, \theta) = \mathbb{E}_{\tau_n^\theta \sim P_n(.|\theta), \tau_n^{\theta,\theta_n} \sim P_n(.|\theta,\theta_n)} \left[ L_n(\theta_n, \theta) \nabla_\theta \log \pi_{\theta_n}(\tau_n^{\theta,\theta_n}) + L_n(\theta_n, \theta) \nabla_{\theta_n} \log \pi_\theta(\tau_n^\theta) \right] \tag{24}$$

## B  CONNECTION TO META-LEARNING

We observe that there exists a natural connection between our proposed distributed learning and gradient based meta-learning techniques such as the one used in [23,24]. We briefly introduce gradient based meta-learning here and draw connections from our work to that of meta-learning.

### B.1 MODEL-AGNOSTIC META LEARNING (MAML)

Consider a series of RL tasks $\mathcal{T}_i$ that one would like to learn. Each task can be thought of as a Markov Decision Process (MDP) $\mathcal{M}(S, A, R, \mathcal{P}')$ consisting of observations $s \in S$, actions $a \in A$, a state transition function $\mathcal{P}'(s_{t+1}|s_t, a_t)$ and a reward function $R(s_t, a_t)$. To solve the MDP (for each task), one would like to learn a policy $\pi : s \to a$ that maximizes the expected sum of rewards over a finite time horizon $H$, $\max_\pi[\sum_{t=1}^H R_t(s_t, a_t)]$. Let the policy be represented by some function $f_\theta$ where $\theta$ is the initial parameters of the function.

In MAML [24] the authors show that, it is possible to learn a policy $\pi_\theta$ which can be used on a task $\mathcal{T}_i$ to collect a limited number of trajectories $\tau_\theta$ or experience $\mathcal{D}$ and quickly adapt to a task specific policy $\pi_{\theta'_i}$ that minimizes the task specific loss $L_{\mathcal{T}_i}(\tau_\theta) = -\mathbb{E}_{s_t, a_t \sim \tau_\theta}[\sum_{t=1}^H R_t(s_t, a_t)]$. MAML learns task specific policy $\pi_{\theta'_i}$ by taking the gradient of $L_{\mathcal{T}_i}(\tau_\theta)$ w.r.t $\theta$. This is then followed by collecting new trajectories $\tau_{\theta'_i}$ or experience set $\mathcal{D}'_i$ using $\pi_{\theta'_i}$ in task $\mathcal{T}_i$. $\theta$ is then updated by taking the gradient of $L_{\mathcal{T}_i}(\tau_{\theta'_i})$ w.r.t $\theta$ over all tasks. The update equations for $\theta'$ and $\theta$ are given as:

$$\theta'_i := \theta - \alpha \nabla_\theta L_{\mathcal{T}_i}(\tau_\theta), \qquad \theta := \theta - \beta \nabla_\theta \sum_{\mathcal{T}_i} L_{\mathcal{T}_i}(\tau_{\theta'_i}) \qquad (25)$$

where $\alpha$ and $\beta$ are the hyperparameters for step size. Authors in [23] extend MAML to show that one can think about MAML from a probabilistic perspective where all tasks, trajectories and policies can be thought as random variables and $\theta'$ is generated from some conditional distribution $P(\theta'|\theta, \tau_\theta)$.

### B.2 DISTRIBUTED OPTIMIZATION FOR MULTI AGENT SYSTEMS

We observe the meta-policy $\pi_\theta$ that MAML attempts to learn and uses as an initialization point for the different tasks is similar in spirit to the central policy $\theta$ DIMAPG attempts to learn and execute on all agents. In both, approaches $\theta$ captures information across multiple tasks or multiple agents. An important difference between our work and MAML or meta-learning is that during execution (post training) we execute $\theta$ while MAML uses $\theta$ to do a 1-shot adaptation for task $\mathcal{T}_i$ and then executes $\theta'_i$ on $\mathcal{T}_i$.

Another interesting point to note here is the difference in the trajectories $\tau_{\theta'_i}$ that is used by MAML and the trajectory $\tau_n^{\theta, \theta_n}$ that is used by DIMAPG to update task or agent specific policy $\theta'_i$ or $\theta_n$. In the distributed optimization for multi-agent setting, due to the non-stationarity, it is absolutely necessary that we ensure the other agents are held constant (to $\theta$) while agent $n$ is optimizing its task specific policy $\theta_n$. MAML has no such requirement.

## C EXPERIMENTAL DETAILS

### C.1 A3C KITCHENSINK AND TRPO KITCHENSINK

For A3C KitchenSink, we input the agents observation and reshape it into a $n \times m$ matrix. This is then fed into a 2D convolution layer with 16 outputs, Elu activation and a kernel size of 2, stride of 1. The output from this layer is fed into another 2D convolution layer with 32 outputs, Elu activation and a kernel size of 2, stride of 1. The output from this layer is flattened and fed into a fully connected layer with 256 outputs and Elu activation. This is followed by feeding into a LSTM layer with 256 hidden units. The output from the LSTM is then fed into two separate fully connected layers to get the policy estimate and the value function estimate. Actor-critic loss is setup and minimzied using Adam with learning rate 1e-4. For TRPO Kitchensink, we setup similar policy layer and value function layer.

### C.2 DIMAPG

For this task, we used a neural network policy with two hidden layers with 100 units each. The network uses a ReLU non-linearity. Depending on the experiment we compute agent specific gradient updates using REINFORCE and TRPO for the central policy gradient updates. The baseline is fitted separately at each iteration for all agents sampled from the population. We use the standard linear feature baseline. The learning rate for agent specific policy updates $\alpha_1 = \alpha_2 = 0.01$. Learning rate for

central policy updates $\epsilon = 0.05$. In practice, to adapt $\theta$ to $\theta_n$ we do multiple gradient steps. We observe k=3 (number of gradient steps) is a good choice for most tasks. For both $\theta$ and $\theta_n$ updates, we collect 25 trajectories.

## C.3 PREDATOR PREY

For the predator prey experiment, we would also like to report additional results showing that after termination, the DIMAPG agents are able to learn faster than the agents trained using other baselines.

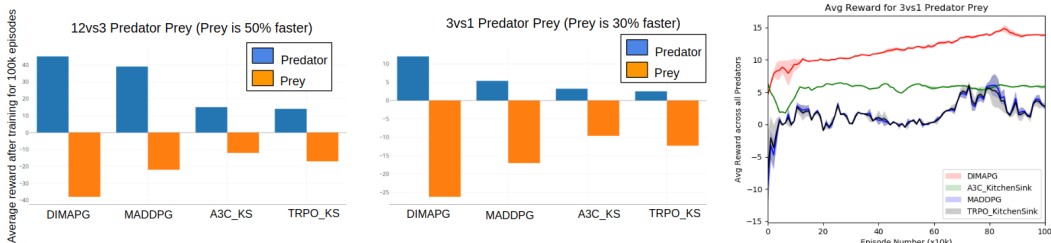

Figure 5: **Performance on Predator Prey populations** In this setting we learn two sets of policies, one for the predators and one for the prey. Here, we observe that after training for a fixed number of episodes, proposed DIMAPG algorithm is able to learn faster than the other algorithms.

## C.4 SURVIVOR

In this experiment, the environment is populated with agents and food particles. The agents must learn to survive by eating food. To do so they can either rush to gather food and get reward or monopolize the food by first killing other agents (killing other agents results in a small negative reward). Each agent in this environment also has orientation. The agents can either chose to one of 12 neighboring cells or stay as is, or chose to attack any agent or entity in 8 neighboring cells. Finally the agent can also choose to turn right or left. At every step, the agents receive a "step reward" of -0.01. If the agent dies, its given a reward of -1. If the agent attacks another agent, it receives a penalty of -0.1. However, if it chooses to attack another agent by forming a group it receives an award of 1. The agent also gets a reward of +5 for eating food.

As stated in the main paper, it is observed that in the case when $N = 225$, the agents do not kill each other and instead learn to gather food. When the number of agents is increased to $N = 630$ agents close to the food rush in to gather food while those further away start killing other agents. We present a snapshot of the learned policy in Figure 1 and Figure 2.

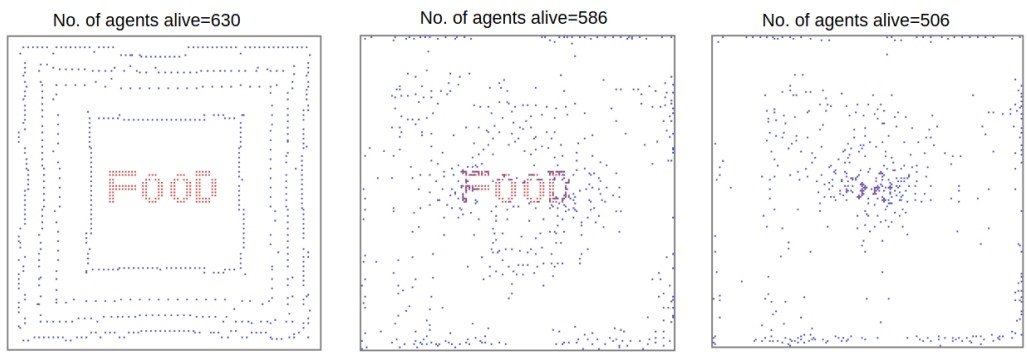

Figure 6: **Learned policy on Survivor(N=230)** When the number of agents is small, agents prefer to eat food instead of killing each other. Most agents survive in this setting.

No. of agents alive=230        No. of agents alive=230        No. of agents alive=226

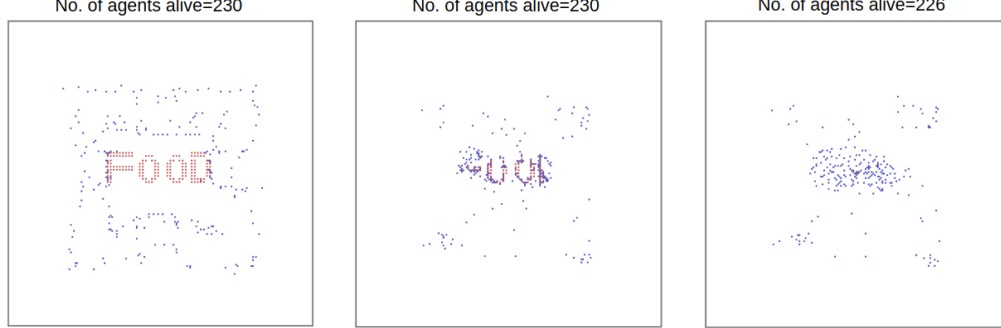

Figure 7: **Learned policy on Survivor(N=630)** When the number of agents is much larger than the amount of food in the environment, the agents closer to the food rush in to gather food. We observe that the agents further away (near the walls) form teams and try to take down other agents thus maximizing reward for the group. This can also be interpreted as follows: Agents who can observe the food within their sensing range choose to rush in food. Agents who do not observe food within their sensing range choose to form groups to take down other agents.

