# OpenReview forum: "COLLABORATIVE MULTIAGENT REINFORCEMENT LEARNING  IN HOMOGENEOUS SWARMS"
_ICLR.cc/2019/Conference_

### Official Review · AnonReviewer1 · 2018-11-01
**Interesting paper, some concerns with formalism and objective. Missing baselines.**

**Rating:** 5
**Confidence:** 4

**Review:**

This paper proposes a distributed policy gradient method for learning policies with large, collaborative, homogeneous swarms of agents.

Formalism / objective:
The setting is introduced as a "collaborative Markov team", so the objective is to maximise total team reward, as expressed in equation (3). This definition of the objective seems inconsistent with the one provided at line (14): Here the objective is stated as maximising the agent's return, L_n, after [k] steps of the agent updating their parameters with respect to L_n, assuming all other agents are static. I think the clearest presentation of the paper is to think about the algorithm in terms of meta-learning, so I will call this part the 'inner loop' from now on.
Note (14) is a very different objective: It is maximising the return of an agent optimising 'selfishly' for [k] steps, rather than the "collaborative objective" mentioned above. This seems to break with the entire premise of collaborative optimisation, as it was stated above.
My concern is that this also is reflected in the experimental results: In the food gathering game, since killing other agents incurs "a small negative reward", it is never in the interest of the team to kill other team-mates. However, when the return of individual agents is maximised both in the inner loop and the outer loop, it is unsurprising that this kind of behaviour can emerge. Please let me know if I am missing something here.

Other comments:
-The L_n(theta, theta_n) is defined and used inconsistently. Eg. compare line (9), L_n(theta_n, theta), with line below, L_n(theta, theta_n). This is rather confusing
-In equation (10) please specific which function dependencies are assumed to be kept? My understanding is that \theata_n is treated as a function of theta including all the dependencies on the policies of other agents in the environment?
-Related to above, log( pi_\theta_n ( \tan_n)) in line 16 is a function of all agents policies through the joint dependency on \theta. Doesn't that make this term extremely expensive to evaluate?
-Why were the TRPO_kitchensink and A3C_kitchensink set up to operate on the minimum reward rather than the team reward as it is defined in the original objective? It is entirely possible that the minimum reward is much harder to optimise, since feedback will be sparse.
-The survival game uses a discrete action space. I am entirely missing MARL baseline methods that are tailored to this setting, eg. VDN, QMIX, COMA etc to name a few. Even IQL has not been tried. Note that MADDPG assumes a continuous action space, with the gumble softmax being a common workaround for discrete action spaces which has not been shown to be competitive compared to the algorithms mentioned above.
-Algorithmically the method looks a lot like "Learning with Opponent Learning Awareness", with the caveat that the return is optimised after one step of 'self-learning' by each agent rather than after a step of 'Opponent-learning'. Can you please elaborate on the similarity / difference?
-Equation (6) and C1 are presented as contributions. This is the standard objective that's commonly optimised in MARL when using parameter sharing across agents.

---

### Official Review · AnonReviewer3 · 2018-11-02
**This paper addresses MARL in the case of having many homogeneous agents with shared policy. Some concerns about the the validity of projection.**

**Rating:** 4
**Confidence:** 4

**Review:**

The introduction of the paper is well-written and the authors quite clearly explain the purpose; however, I believe that the notation should be revisited to further simplify them. The algorithm pretty much is similar to the A2C algorithm (very minor differences) and overall, I don't see the contribution of the paper to be significant enough. Also, there are a few other concerns that I summarize next:

1) In example 1, just knowing the relative distance with all other agents is not equivalent to knowing the full state of the environment. This is because the angles with the other agents are important; i.e. you need to know (r,\theta) Polar Coordinates.

2) I personally don't like using the word "constrained'' used in this paper. Going back to constrained RL literature, the purpose of constrained RL is, for example, not entering the hazardous states. At the first time reading the paper, I thought that the constraints are referring to such cases, e.g. make sure that the agents never hit each other. But, the concept of constraint used in this work is totally different and it simply means copying the network weights.

3) In section 3, using the neural networks and averaging of the weight does not make any sense. What does it mean to average weights of several NN? NN is a nonlinear function approximator, and you cannot average weights. Based on your algorithm, I see that you aggregate the gradients which is a correct approach. In fact, the projection step defined in page 5 is never used in your implementation I guess, because otherwise, this algorithm will not work.

4) The distributed model pretty much resembles A2C algorithm where each agent can be considered as a thread. At every time, you only do a gradient step in one of the threads and for the rest, you use the central policy. This way, you stabilize the non-stationarity caused by concurrently learning policy. I do not see any major difference.

5) What is the reason that you do not use the Critic?

6) Having $\theta_n=\theta$ implies that $\pi_n = \pi$, but the other way around does not hold. Constraints of (8) are not equivalent to (5).

7) Are you using different policies for different agents when using MADDPG or TRPO_Kitchensink? I think for a fair comparison, the agents should also share the policies in these algorithms too. It is very hard to believe that TRPO_Kitchensink and MADDPG almost learn nothing, or even learn in reverse direction (Fig 3).

8) I think that the baseline comparisons for the case of having a small number of agents are necessary.
Minor:
* In section 3.1, the notations are over-populated. I would suggest simplifying the notations.
In (4), (5),(6), argmax_\pi
* (16) is simply the sum of gradients of two consecutive policy gradient steps which can be derived by (sum of grad = grad of sum). You might add this as an intuition beyond this formula.

---

### Official Review · AnonReviewer2 · 2018-11-03
**An interesting idea, with some good theoretical justification and interesting evaluations. There could be more precise details (particularly on experiments) and the approach used to combine gradients could be better related to prior work.**

**Rating:** 6
**Confidence:** 3

**Review:**

## Summary
The authors present an approach to training collaborative swarms of agents based around giving all agents identical (or near identical) policies. The training regime involves individual agents rolling out trajectories based on slight perturbations of an agent of focus keeping the policy of other agents fixed. This is repeated for each agent, then these trajectories are used to batch update the joint policy with an average gradient.

On the whole, I think the paper is well written and the idea novel. There are places where the explanations could be clearer and details more explicit (see below for examples). There are some interesting evaluations but I am not sure these are as rigourous as they could be, in particular (but not limited to) the survival game. I am, however, recommending this for acceptance as on balance the positives outweigh the negatives.

## More detailed comments
The authors could make it a bit clearer what existing work on averaging policy gradients exist, and whether their approach is a natural extention of these existing approaches to their swarm domain, or whether there is additional novelty there. It is unclear to me which is the case. They talk about meta-learning in the related work but it is unclear precisely how they relate this to their own work.

The authors could describe their experiments a little more explicitly. For instance, they say that agents are penalised for getting too close in the navigation task, but do not say how this penalty is constructed. Is it a step function based on distance or something else? Also, they should state what parameters they use for each of the environmental factors, e..g minimum distance etc.

The survival game is poorly described, as are choices for the evaluation of it. I realise these games are designed elsewhere, but if the exact same parameters are used as in the original papers then this should be stated. Finally, I am a little unclear why the survival game cannot be compared with other algorithms, even if those algorithms fail to learn anything. I realise that the algorithms with decentralised actors won't scale here, but something like the mean field approaches described by the authors in the related work, or even less sophisticated algorithms using some (but not all) features of their own approach would show something interesting. And the choices of 225 and 630 agents needs better justification.

---

### Meta-Review · Area_Chair1 · 2018-12-14

**Confidence:** 4
**Recommendation:** Reject

**Metareview:**

Pros:
- interesting novel formulation of policy learning in homogeneous swarms
- multi-stage learning process that trades off diversity and consistency (fig 1)

Cons:
- implausible mechanisms like averaging weights of multiple networks
- minor novelty
- missing ablations of which aspect is crucial
- dubious baseline results
- no rebuttal

One reviewer out of three would have accepted the paper, the other two have major concerns. Unfortunately the authors did not revise the paper or engage with the reviewers to clear up these points, so as it stand the paper should be rejected.